# Reference Intervals for Platelet Counts in the Elderly: Results from the Prospective SENIORLAB Study

**DOI:** 10.3390/jcm9092856

**Published:** 2020-09-03

**Authors:** Wolfgang Hermann, Lorenz Risch, Chris Grebhardt, Urs E. Nydegger, Benjamin Sakem, Mauro Imperiali, Harald Renz, Martin Risch

**Affiliations:** 1Faculty of medical sciences, Private Universität im Fürstentum Liechtenstein, Dorfstrasse 24, 9495 Triesen, Liechtenstein; whermann86@gmail.com (W.H.); lorenz.risch@risch.ch (L.R.); 2Labormedizinisches Zentrum Dr. Risch, Wuhrstrasse 14, 9490 Vaduz, Liechtenstein; 3Center of Laboratory Medicine, University Institute of Clinical Chemistry, University of Bern, 3012 Bern, Switzerland; 4Faculty of Medicine, University of Basel, Klingelbergstrasse 61, 4056 Basel, Switzerland; chris.grebhardt@stud.unibas.ch; 5Labormedizinisches zentrum Dr. Risch, Waldeggstrasse 37, 3097 Liebefeld, Switzerland; urs.nydegger@risch.ch (U.E.N.); benjamin.sakem@risch.ch (B.S.); 6Centro Medicina di Laboratorio Dr. Risch, Via Arbostra 2, 6963 Pregassona, Switzerland; mauro.imperiali@risch.ch; 7Institute of Laboratory Medicine and Pathobiochemistry, Molecular Diagnostics, Philipps University Marburg, University Hospital Giessen and Marburg, Baldingerstraße, 35043 Marburg, Germany; Harald.Renz@uk-gm.de; 8Zentrallabor, Kantonsspital Graubünden, Loësstrasse 170, 7000 Chur, Switzerland

**Keywords:** elderly, female, male, platelet count, population-based setting, prevalence, reference intervals, seniors, sensitivity, specificity

## Abstract

Currently, age- and sex-independent reference limits (RLs) are frequently used to interpret platelet counts in seniors. We aimed to define and validate reference intervals (RIs) for platelet counts within the framework of the prospective SENIORLAB study. Subjectively healthy Swiss individuals aged 60 years and older were prospectively included and followed for morbidity and mortality. Participants who had circumstances known to affect platelet counts were excluded. The obtained RIs were validated with indirect statistical methods. Frequencies of abnormal platelet counts in a population-based setting, including 41.5% of the entire age-specific population of the Principality of Liechtenstein, were compared by using age- and sex-independent RIs and the RLs obtained in the present study. For males (*n* = 542), 95% RIs for platelet counts were defined as follows: 150–300 × 10^9^/L (60–69 years); 130–300 × 10^9^/L (70–79 years); and 120–300 × 10^9^/L (80 years and above). For females (*n* = 661), the consolidated age-independent 95% RI was 165–355 × 10^9^/L. These RI values were validated by indirect RI determination of 51,687 (30,392 female/21,295 male) patients of the same age. Age- and sex-independent RIs exhibited imbalanced frequencies of abnormal platelet counts between the two sexes, which were corrected by introducing age- and sex-specific RLs. In conclusion, females have higher platelet counts than males. Whereas the upper RL for males remains constant, the lower RL decreases with age. We propose to abandon the practice of employing sex- and age-independent RL for platelet counts in the elderly.

## 1. Introduction

Platelets, the smallest circulating blood cells, are produced in the bone marrow and play a crucial role in hemostasis, but also have extrahemostatic functions [1,2]. There are many reasons for altered platelet counts, which can manifest as either thrombocytopenia or thrombocytosis [3,4,5]. Thrombocytopenia or thrombocytosis can occur as an imbalance between platelet production in the bone marrow, consumption in the peripheral tissues, and platelet distribution within the organs [3,6]. Whereas no circumstances are known to increase platelet lifespan (in contrast to, e.g., red blood cells), the lifespan of platelets is shorter in conditions with increased platelet consumption [3].

Primary thrombocytosis is caused by myeloproliferative neoplasia and other bone marrow diseases, whereas secondary thrombocytosis can occur due to metabolic disorders (e.g., iron deficiency), infections and chronic inflammation (e.g., tuberculosis, chronic inflammatory bowel disease), posttraumatic disorders (e.g., surgery with considerable blood loss, injuries), or hyposplenism, or in the context of paraneoplastic disease in patients with metastatic cancer, or in regenerative thrombocytosis (e.g., after chemotherapy, hemolytic disease) [3].

Thrombocytopenia can occur due to increased consumption, e.g., in the context of adverse drug reactions, immune-mediated thrombocytopenia (e.g., ITP or posttransfusion purpura), thrombotic microangiopathy (e.g., hemolytic-uremic syndrome, thrombotic thrombocytopenic purpura, HELLP (haemolysis, elevated liver enzyme levels, low platelet count) syndrome), secondary to underlying disease (e.g., lupus erythematosus, lymphoma, malaria), or in mechanic heart valves [3]. Other mechanisms lead to decreased production in genetic disease, viral infection (e.g., Parvovirus, human immunodeficiency virus, cytomegalovirus), bone marrow damage (e.g., in cytostatic therapy, radiation, immunosuppression), vitamin deficiency (e.g., vitamin B12, folic acid), bone marrow infiltration (e.g., in leukemia, lymphoma, carcinoma), and aplastic anemia as well as abnormal distribution, such as in splenomegaly or pseudothrombocytopenia due to dilution in pregnancy [3,7]. Together, platelet count abnormalities are a symptom rather than a disease and can be observed in a wide range of medical and surgical conditions, some of which have high frequency among seniors. Abnormal platelet counts therefore can be regarded as an early indicator of undetected occult disease or a predictor of outcomes in established disease [8,9,10,11].

Platelet counts are mainly ordered for two reasons. First, the parameter is determined to assess bleeding risk depending on underlying disease and other factors [12]. In immune thrombocytopenia, platelet counts less than 20 or 30 × 10^9^/L have been suggested to be associated with an increased risk of bleeding [13,14]. Second, platelet counts below or above the reference limits can indicate underlying disease. If these reference limits are inaccurate, the diagnosis of underlying disease may be overlooked, with potential negative sequelae for patients (e.g., delayed diagnosis with delayed therapy). In order to take decisions as accurately as possible, decision limits (defined as upper and lower limits of the reference interval) for diagnosing thrombocytopenia and thrombocytosis have to be as accurate as possible. Regarding platelet counts, there is no clear position on whether platelet count reference intervals should be age- and sex-specific. Whereas some authors propose stratification according to sex, age, or other characteristics [15,16,17,18,19,20], sex- and age-independent reference intervals are commonly used in clinical practice [21,22,23,24,25]. The importance of having accurate reference intervals for platelet counts available has been evident for a long time [26]. Since conditions associated with thrombocytopenia or thrombocytosis occur relatively frequently in seniors and since reference intervals for seniors are generally not well described, we determined reference intervals for platelet counts in seniors within the framework of the SENIORLAB study [27]. This was done because there is a lack of data on patients older than 70 years and investigated groups were rather small. Such an approach determining reference intervals in a study, where study participants have been recruited for this purpose technically is called a direct method of reference interval determination [28]. We further validated our findings with another statistical approach in an independent setting of clinical patients. An approach determining reference intervals in a study, where reference intervals are estimated from patient results technically is called an indirect method of reference interval determination [28]. Finally, we compared the frequency of abnormal platelet counts by applying modified and conventional platelet count reference intervals in a population-based setting.

## 2. Materials and Methods

### 2.1. Study Participants and Patient Cohorts

The present analysis was done within the framework of the SENIORLAB study, a prospective cohort study for which the detailed study protocol has been published elsewhere [27]. In brief, the SENIORLAB study comprises 1467 subjectively healthy individuals who were seen between May 2009 and December 2011 for a baseline visit to obtain a detailed history, anthropometric data, and blood by venipuncture under optimal preanalytical conditions. From December 2013 to December 2014, the study participants were followed up regarding mortality and morbidity, with a 100% follow-up rate. For the present analysis, we included participants age 60 years and older who resided in Switzerland and were in a fasting state at the baseline examination. A further inclusion criterion, as proposed earlier as an objective indicator of health in seniors, was survival. Because life expectancy in seniors differs according to age, we deliberately chose the following minimal survival criteria for exclusion from the present analysis: death before first follow-up for participants < 80 years of age, death within at least 3 years between age 80 and 85, death within at least 2 years between age 85 and 90, and death within at least 1 year for age 90 and older. Further exclusion criteria were polypharmacy (defined as the use of more than 5 pharmacologically active compounds) and circumstances known to affect platelet counts: ferritin < 15 μg/L (as an indicator of iron deficiency) [29], vitamin B12 concentration < 130 pmol/L (as an indicator of vitamin B12 deficiency) [30] or red blood cell (RBC) folate < 340 nmol/L (as an indicator of folate deficiency) [31], and c-reactive protein (CRP) > 8.7 mg/L or white blood cell count > 10.4 × 10^9^/L (as indicators of inflammation) [20]. The participants provided written informed consent to participate in the study. The study is in accordance with the Declaration of Helsinki and was approved by the local ethics committee (KEK Bern, Switzerland; 166/08). The study is registered in the International Standard Randomized Controlled Trial Number registry (ISRCTN53778569).

To validate the reference intervals from the SENIORLAB study, an indirect determination of reference intervals was obtained from results determined in clinical patients. For this purpose, routine results from the Dr. Risch Medical Laboratory Center were employed. This laboratory serves physicians in offices and, to a lesser extent, hospitals. We included platelet data from samples with concurrently available information on hemoglobin concentrations and white blood cell counts obtained from 3 September 2004 through 15 July 2020. In order to minimize contamination of the non-diseased subgroup by samples from diseased individuals [32], samples with hemoglobin concentration (as a crude indicator of anemia due to vitamin or iron deficiency bone marrow disorder) less than 130 g/L or more than 170 g/L (for males) or less than 120 g/L or more than 160 g/L (for females) and white blood cell counts more than 10.4 or less than 3.8 (as a crude indicator of inflammation) were excluded from this analysis [20,33]. The last available result per patient (i.e., only one result per patient) was taken for this analysis. The responsible ethics committee (KEK Bern, Switzerland; BASEC 2020-00139) approved this validation of directly evaluated reference intervals by means of indirect methods employing routine laboratory measurements and waived the need for informed consent

Commonly employed platelet count reference intervals for seniors do not differ according to age and sex. It is therefore of interest whether novel reference intervals would lead to different frequencies of pathological results. For this purpose, we analyzed anonymized platelet count results obtained from a small country with just one central laboratory, the Principality of Liechtenstein [34,35]. During the period from 1 January 2013 to 31 December 2019, a total of 12,897 individuals aged 60 or older (corrected for 1613 deaths) were living in the country [35]. Of these, 5351 individuals (2593 males, 2758 females; 41.5%, 95% confidence interval (CI) [40.6, 42.3]) of the entire population had at least one platelet count available, with a total of 22,800 determinations available. The protocol for this part of the study was verified by the responsible ethics committee (KEK Zürich, Zürich, Switzerland; BASEC 2020-00918) and informed consent was waived.

### 2.2. Laboratory Methods

Platelet determinations for direct reference intervals were done on a Sysmex XE 5000. In our experience, the coefficient of variation as a measure of imprecise platelet counts has been found to be 0.9–1.8%, depending on the mean platelet concentration [27]. Vitamin B12, folate, RBC folate, ferritin, and CRP values were determined for exclusion of study participants with vitamin or iron deficiency or inflammation, as described elsewhere [27,36]. The platelet measurements for the indirect determination of reference intervals were done with several hematology analyzers from Sysmex (XT 2100, XE 5000, XS 1000, XN 1000; Sysmex, Horgen, Switzerland) and Horiba (Abx Pentra, Micros; Axon Lab, Baden, Switzerland).

### 2.3. Statistical Methods

Continuous variables were summarized by means of median and interquartile range (IQR). Assessment for Gaussian distribution was done by visual inspection of results. Proportions are given as percentages together with their 95% confidence intervals. Rank correlations were assessed with Kendall’s tau, whereas comparisons among 3 groups were done with the Kruskal–Wallis test. Linear trends among 3 groups (e.g., stratified age groups, i.e., 60–69 years, 70–79 years, and ≥80 years) were assessed with the Jonckheere-Terpstra test. Direct evaluation of reference intervals was done according to Clinical Laboratory Standards Institute (CLSI) guideline EP28-A3c [28]. For direct evaluation of reference intervals, we decided on partitions after visually inspecting scatter plots for age-related trends as described by Altman [37]. Partitions were statistically evaluated with the Harris and Boyd method [28]. Double-sided robust reference intervals with their 90% confidence intervals were calculated. Variables were transformed by the Box–Cox method to achieve symmetrical distribution of the data. We eliminated outliers according to Tukey. Finally, we checked whether enough samples were available for calculation of reference intervals by checking whether the 90% CI was less than 0.2 of the 95% reference interval [28].

We then validated the obtained direct reference intervals by calculating indirect reference intervals from a routine group by means of Reference Limit Estimator software [38,39,40,41,42,43]. This software works on a Microsoft Excel interface and uses R for statistical calculations. It was released by the decision limits working group of the German Association of Clinical Chemistry and Laboratory Medicine (DGKL) and has been used in several investigations of indirect evaluation of reference intervals [32,42,43,44,45,46]. In brief, this software estimated a smoothed kernel density function, which resulted in a dataset consisting of 3 subgroups, the non-diseased “healthy” population and 2 subgroups of supposedly “diseased” individuals with values lower and higher than those of the healthy subgroup. The samples of the “healthy” population were modeled using a Box–Cox transformed truncated normal distribution in order to transform non-normal distributions into symmetrical distributions. The parameters for the Box–Cox transformation and the truncated normal distribution were estimated using the maximum likelihood method, leading to parametrical distributions of “healthy” samples. From this distribution, the 2.5th and 97.5th percentiles were estimated together with the permissible uncertainty of the upper and lower reference limits [42]. Concordance of results from direct and indirect reference interval determination was compared by checking whether the range of permissible uncertainty of the upper and lower reference limits in the indirect determination overlapped with the 90% CI of the lower and upper limits in the direct determination.

Finally, the potential impact of applying the evaluated novel reference intervals for platelet counts was investigated in a population-based setting. For this, the frequency of pathological results (i.e., thrombocytopenia and thrombocytosis) was assessed by comparing commonly employed sex- and age-independent reference intervals (150–350 × 10^9^/µL) [21] to the frequency of pathological results employing the novel sex- and age-dependent reference intervals in the results obtained from patients in the Principality of Liechtenstein. Direct reference intervals were calculated with MedCalc version 17.4 (MedCalc Software bvba, Ostend, Belgium; http://www.medcalc.org; 2017). Graphs were drawn using Prism 5.04 (GraphPad Software, Alameda, CA, USA) or MedCalc.

## 3. Results

### 3.1. Direct Evaluation of Reference Intervals

After applying the exclusion criteria according to the participant inclusion chart in Figure 1, a total of 1203 individuals were available for statistical analysis. Baseline characteristics of the study participants are given in Table 1. As can be seen in Figure 2, there were no age-related changes of platelet counts in females, whereas in males, the lower limit appeared to decline and the upper limit appeared to remain constant. Females had significantly higher platelet counts than males (*p* < 0.001). In both females (τ = −0.05; *p* = 0.05) and males (τ = −0.05; *p* = 0.06), there was a weak correlation between platelet count and age, but with marginal significance. In order to have sufficient individuals per reference group to provide solid reference intervals, we chose to initially partition by decade (i.e., 60 to 69 years, 70 to 79 years, 80 years and older) and sex, resulting in three reference intervals each for females and males (Table 2). Platelet counts in the three female groups did not differ significantly. We therefore collapsed the platelet count reference interval for females to a common reference interval of 167 (90% CI (164,170)) to 355 (90% CI (347, 362)). However, males showed significant differences among the three age groups (*p* = 0.02) with a significant negative trend (*p* = 0.006). As can be seen in Table 2, for males, the lower reference limit decreases by about 20% with age, whereas the upper reference limit remains constant. According to Harris and Boyd, the number of reference individuals was sufficient in each of the investigated subgroups in order to have an acceptably narrow 90% CI compared to the double-sided 95% reference interval. Putting this together, we determined the following platelet count reference intervals for seniors aged 60 years and older: 165–355 × 10^9^/L for females, and 150–300 × 10^9^/L (60–69 years), 130–300 (70–79 years), and 120–300 (80 years and above) for males.

### 3.2. Validating the Reference Intervals with Indirect Methods

Routine data from a total of 113,877 samples originating from 51,687 patients (21,295 male, 30,392 female) aged 60 years or older were available for indirect evaluation of reference intervals. The reference intervals are shown in Table 3. It can be seen that the lower and upper reference limits for females and the upper reference limit for males largely remain unchanged, whereas the lower limit of the reference interval for males decreases from the 60–69 age range to the 80 and older range by about 17%. This parallels the findings in the direct reference interval determination. It can also be seen that the permissible uncertainty range of the lower reference limit for males aged 60 to 69 years and that of the upper reference limit for females aged 70–79 years differed slightly from the 90% CI in the direct reference interval determination. These differences of the two ranges were, however, small, at 3 × 10^9^/L for males aged 60–69 years and 7 × 10^9^/L for females aged 70–79 years. In all other age and gender strata, the permissible uncertainty range and 90% CI overlap. In the female groups, the collapsed reference range in the indirect reference range determination overlaps with the 90% CI of the direct reference range determination. Together, the findings of the indirect and direct reference interval determinations are corroborated.

### 3.3. Impact of Novel Reference Intervals on Frequency of Abnormal Platelet Counts

In addition to the conventional age- and sex-independent platelet count reference intervals, we applied the three age-stratified reference intervals for males and females and all seniors aged 60 and older. The distribution of platelet counts in males and females in Liechtenstein is shown in Figure 3. In this population-based sample, we determined the prevalence of thrombocytopenic and thrombocytotic samples according to the conventional criteria (150–350 × 10^9^/L), as shown in Table 4 We further determined their prevalence by employing the reference intervals as evaluated in the present study: 165–355 × 10^9^/L for females, and 150–300 × 10^9^/L (60–69 years), 130–300 (70–79 years), and 120–300 (80 years and above) for males (Table 5). The total proportion of abnormal platelet counts was 22.1% (95% CI (21.6, 22.7); 5048/22,800) when using the conventional reference limits and 25.1% (95% CI (24.5, 25.6); 5716/22,800) when using sex- and age-differentiated reference limits. It can be seen in Table 4 that with the conventional reference values, even though in the total proportion of abnormal platelet counts there is only a 1.7% difference between males and females, males had substantially more thrombocytopenia and less thrombocytosis. When applying the above-mentioned new reference limits, the frequency of abnormal platelet counts exhibited a 1.2% difference (Table 5). Furthermore, there was less difference in thrombocytopenia and thrombocytosis frequency in males and females, with thrombocytosis being more frequent in both sexes. Appendix A displays cases that are classified differently as normal result when applying either the conventional or the age- and sex-stratified reference intervals. When employing the conventional lower reference limit, an excess diagnosis of thrombocytopenia can be assumed in 4% (95% CI (3.7, 4.4); 465/11,601) of all investigated samples from male seniors, when compared to the respective age-stratified lower reference limit. 

## 4. Discussion

Sex- and age-independent platelet counts are still often employed in clinical medicine and are recommended [3,21,22,47,48]. Given the importance of the parameter in clinical medicine, this is somewhat interesting, as our study and others could identify sex- and age-specific differences [20,49,50,51]. In seniors, we demonstrate that females have higher reference intervals than males, and that in males, reference limits change with age. Our reference limits were validated with an indirect method in large patient collectives. Further, we demonstrate that the reference limits deviate considerably from commonly used reference limits. In a population-based setting, we finally show that the introduction of age- and sex-stratified reference intervals for seniors leads to a somewhat higher proportion of pathological samples. However, the proportion of thrombocytopenia and thrombocytosis among men and women becomes more balanced.

Reference intervals are important to determine whether a person’s platelet count is normal or not. If the platelet count is abnormal, this could be a first early indicator of potentially treatable disease [10,11]. However, there are also other measures to interpret platelet count values, such as the reference change value (RCV) [52]. Pineda-Tenor et al. showed that, among the 26 parameters investigated in their study, platelet count in seniors was the parameter with the highest index of individuality (II), even if the biological variability was higher than in younger subjects [53]. It is generally accepted that the higher II is above a certain cut-off, the higher the reliability of reference intervals for use in clinical practice [54]. It can therefore be concluded that the determination of reference intervals for platelet counts in seniors is well justified.

The relevant guideline for determination of reference intervals (CLSI EP 28-A3c) proposes two ways of obtaining a reference interval in clinical practice [28]. A laboratory can either adopt already known reference intervals or establish its own reference intervals. Adopting an already known reference interval includes either a small verification study or a subjective assessment of the reference limits by experienced laboratory and clinical staff. These experts inspect whether the known reference interval, which depends on a variety of preanalytical and analytical factors as well as the reference cohort, can be transferred to the clinical environment of the laboratory. Regarding the commonly used reference intervals on platelet counts, a sex- and age-independent reference interval of 150–450 × 10^9^/L can be traced back to a study including 1011 specimens from apparently normal adults [55].

We identified 14 studies and five authoritative sources describing reference intervals for platelet counts in seniors [15,16,17,18,19,20,23,24,25,47,48,49,50,51,55,56,57,58]. One of the studies applied an approach incorporating MPV (mean platelet volume) and sex as defining reference limits [15], whereas most of the other studies used age and/or sex as partitioning factors, as can be seen in Table 6. Only a few of the studies were designed to prospectively include participants with the aim to define reference intervals [18,23,48,50]. Other studies rather included individuals post hoc, e.g., from periodic health exams or population-based cohorts such as NHANES III or others [16,49,51]. Further, a considerable proportion of these mentioned studies did not provide confidence intervals for the estimated reference limits. Although all studies displayed in Table 6 included seniors aged 60 and older, only seven of them included individuals more than 80 years of age [23,24,48,49,50,51,57], and only four covered the whole range from 60 to 100 years [48,49,51,57]. Ten studies provided sex-stratified reference intervals, the vast majority with higher platelet count reference limits for women than men [16,17,18,19,20,48,49,50,51,57]. Among the 12 studies in Table 6, the median lower reference limit for female seniors was 156 × 10^9^/L, which is 8 × 10^9^/L lower than the lower limit of the 90% CI of the lower reference limit in our investigation. The median of the upper reference limit for females is 396 × 10^9^/L is 34 × 10^9^/L higher than the upper limit of the 90% CI of the upper reference in our investigation. To our knowledge, there is so far only one study providing data on age stratification of older men. Nah et al., in their investigation of health examination participants, also described a decline of lower reference limits for males, with a lower limit of 140 × 10^9^/L for the 60–75 age group and 126 × 10^9^/L for those over 75 years, which is also the level found in our investigation applying decade-specific reference intervals [51]. The same study also confirmed the age-independent behavior of the upper reference limit for males [51]. Since the age distributions, sample sizes (i.e., lower sample size with larger CI), ethnicities, and inclusion criteria (e.g., no exclusion of individuals with inflammation or iron deficiency) vary considerably, a comparison among the different studies has limited validity. Nevertheless, we consider that our consolidated reference intervals for females are quite comparable to what is already known from other studies. The present study is so far the only one excluding reference subjects with nutritional deficiencies (e.g., iron, folate, vitamin B12) and objective signs of inflammation, both of which have a considerable influence on platelet counts. This might explain why the reference limits of the present study are higher at the lower reference limit and lower at the higher reference limit than the studies displayed in Table 6. Since the reference range is narrower than in other studies, our reference limits can thus be considered to be more sensitive in detecting underlying disorders.

It is known that platelet counts are determined by several factors and decline with age, and that women have higher platelet counts than men [16,51,59,60,61,62,63,64]. For every 10 years, Santimone et al. reported a sex-adjusted decline of 10 × 10^9^/L [65]. In our healthy cohort, who were prospectively assembled for the definition of reference intervals, we can confirm the significant age-related decline of mean platelet counts in men (216 × 10^9^/L at age 60–69; 210 × 10^9^/L at age 70–79; and 201 × 10^9^/L at age 80 and older) but not in women. This gender difference with an age effect in men but not in women was also seen in the NORIP project [48].

It has been hypothesized that the sex difference in platelet counts after puberty is due to estrogen levels [51]. It could also be hypothesized that in our cohort of postmenopausal women, estrogen levels would be low and would not be the reason for sex differences in platelet counts. Indeed, in our SENIORLAB cohort the median estrogen levels in the women were indeed significantly lower than in the men (<37 pmol/L in women vs. 81 pmol/L in men; *p* < 0.001) [27]. Among the women, 48 were taking hormonal replacement therapy and had significantly higher platelet counts than women without such therapy (median 257 vs. 241 × 10^9^/L; *p* = 0.007). Excluding the women with hormonal replacement therapy did not alter reference limits when compared to the whole cohort, however. Accordingly, our data suggest that estrogen cannot be held accountable for the sex differences in platelet count reference intervals. Further studies are needed to clarify these sex-specific differences.

The reference intervals evaluated and confirmed in the present study differ from commonly employed sex- and age-independent reference intervals, which are recommended by important resources providing reference intervals in hematological parameters [21,47,55,58]. Our study demonstrates that a sex-independent reference interval [21] in a population-based setting leads to an overestimation of thrombocytopenia and an underestimation of thrombocytosis, primarily in men. Unrecognized abnormal platelet counts may lead to delayed diagnosis of underlying disease with its associated burden, whereas false pathological values cause additional psychological burden and cost associated with further investigation [66,67,68]. The application of sex- and age-dependent reference intervals leads to a more balanced frequency of abnormal platelet counts between the two sexes. In order to prevent misdiagnosis in seniors, we therefore recommend the use of age- and sex-specific reference intervals.

Our study has strengths and limitations. We could prospectively assemble a large cohort in order to perform a direct evaluation of reference ranges, the method of choice according the CLSI EP28-A3c [28]. The whole age range from young olds (age 60 to 69 years) to the oldest old (age > 85 years) was taken into account to determine reference intervals in our investigation [69]. Reference limits obtained by direct methods could be validated with an indirect determination of limits in patient samples as well as limits available in the literature. We were able to carefully select reference individuals based on not only anamnestic but also laboratory data, which has rarely (and not as extensively) been done in other studies. We could further assess the impact of introducing modified sex- and age-stratified reference limits compared to using sex- and age-independent limits. The limitation of our investigation is that it cannot be extrapolated to non-Caucasian populations, as ethnicity has been shown to have an impact on platelet counts [16,63,64]. We believe that this limitation does not invalidate our findings.

In conclusion, we identified novel age- and sex-stratified reference intervals for seniors. We can confirm that females have higher platelet counts than males, and the lower reference limit for male seniors decreases with age. We could exclude estrogen as a factor leading to the sex difference in platelet counts in seniors. Introducing age- and sex-specific reference intervals led to a more balanced frequency of thrombocytopenia and thrombocytosis between the sexes. Abandoning the use of sex- and age-independent reference intervals, at least for seniors, seems advisable.

## Figures and Tables

**Figure 1 jcm-09-02856-f001:**
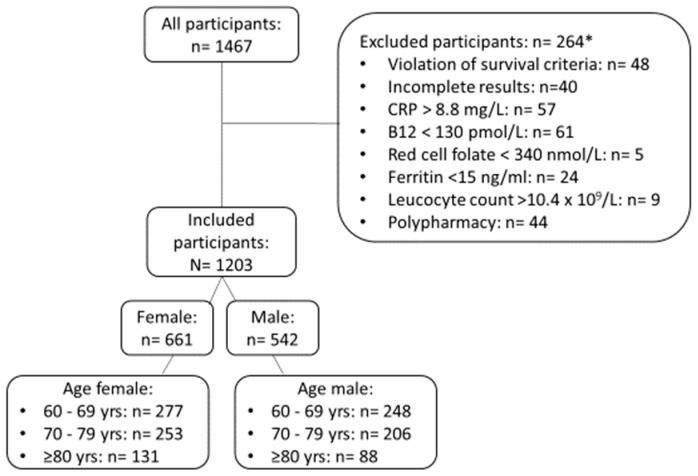
Inclusion algorithm for determination of direct reference intervals for platelet counts in seniors. * One patient could have more than one disorder. CRP = c-reactive protein.

**Figure 2 jcm-09-02856-f002:**
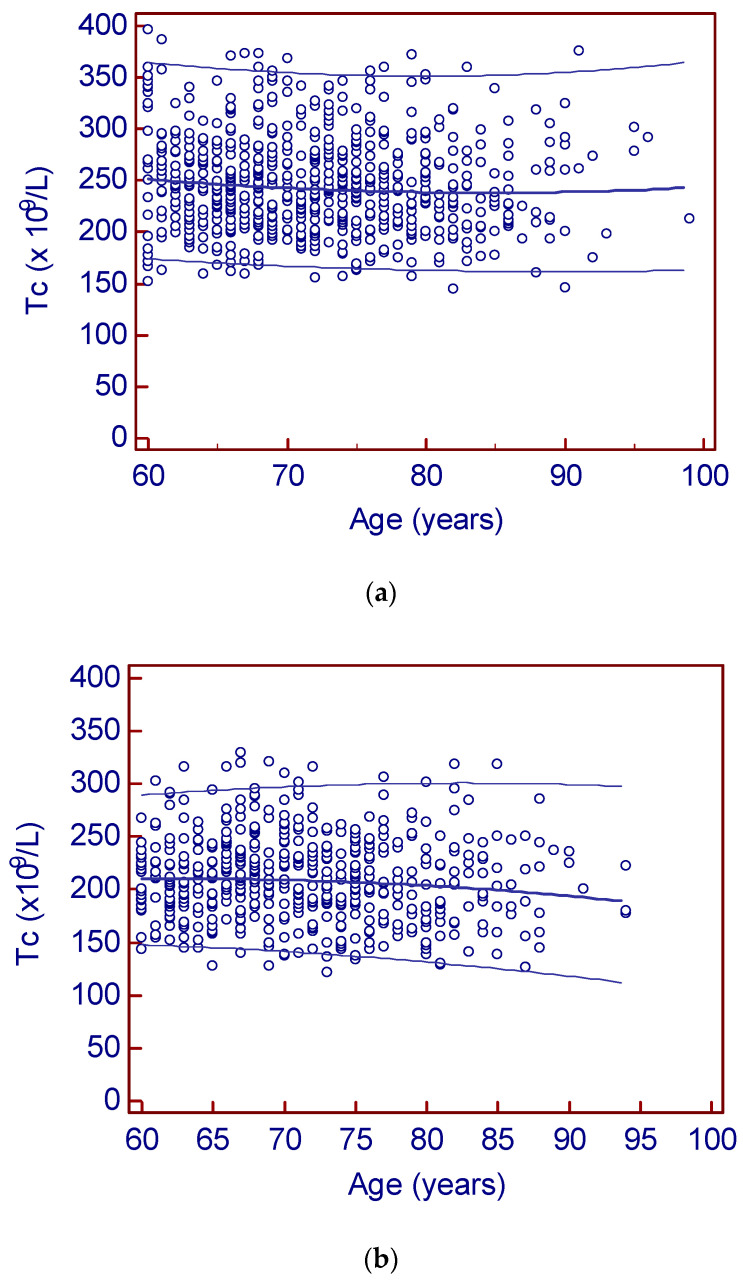
Platelet counts and age stratified according to sex: (**a**) females (**b**) males. The 2.5th to 97.5th percentiles, according to Altman [37], are shown as thinner lines; thicker lines depict medians.

**Figure 3 jcm-09-02856-f003:**
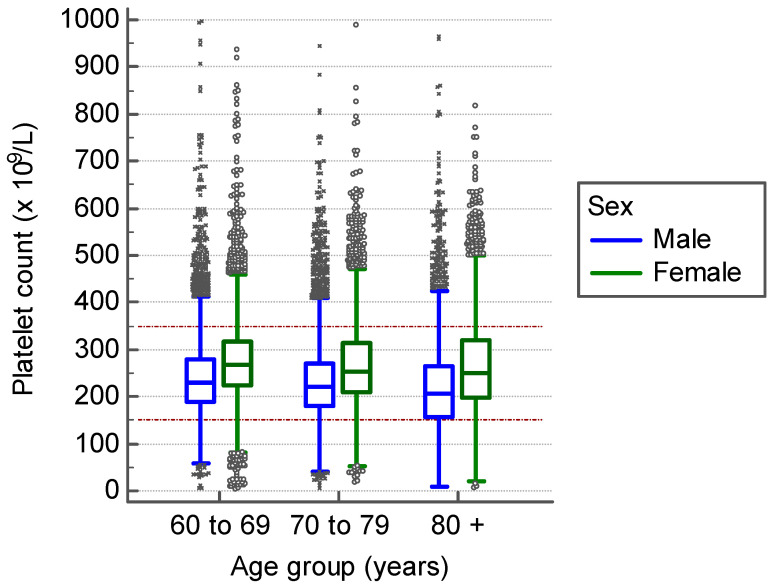
Platelet counts measured in Liechtenstein from 1 January 2013 to 31 December 2019, stratified by age and sex. Dotted lines represent conventionally used sex- and age-independent reference interval (150–350 × 10^9^/L) [21].

**Table 1 jcm-09-02856-t001:** Baseline characteristics of participants included in direct reference interval determination for platelet count. Summary statistics are given as medians with interquartile range (IQR).

Characteristic	Females (*n* = 661)	Males (*n* = 542)
Age, years	72 (66, 78)	70 (65, 76)
Body mass index, kg/m^2^	24.5 (22.1, 27)	25.4 (23.7, 28.1)
Systolic blood pressure, mmHg	146 (130, 162)	146 (132, 162)
Hemoglobin, g/dL	137 (131, 142)	149 (143, 156)
Leukocytes, ×10^9^/L	5.3 (4.5, 6.3)	5.6 (4.8, 6.6)
Platelet count, ×10^9^/L	243 (211, 278)	208 (182, 237)
Reticulocytes, ×10^6^/L	0.04 (0.03, 0.05)	0.04 (0.04, 0.05)
eGFR (CKD-EPI_Creatinine_), mL/min/1.73 m^2^	80 (69, 89)	80 (70, 88)
ALAT, U/L	17 (14, 21)	21 (17, 27)
HbA1_c_, %	5.8 (5.6, 6)	5.8 (5.6, 6.1)
Ferritin, μg/L	90 (59,131)	159 (93,237)
Vitamin B12, pmol/L	248 (198, 322)	234 (192, 289)
Red blood cell folate, nmol/L	950 (701, 1244)	888 (702, 1159)
CRP, mg/L	1.4 (0.7, 2.6)	1.2 (0.7, 2.3)

ALAT = alanine aminotransferase.

**Table 2 jcm-09-02856-t002:** Direct reference intervals for platelet counts stratified according to age and sex. Collapsed reference interval for females without age stratification is 167 (90% CI (164, 170)) to 355 (90% CI (347, 362)). CI, confidence interval.

	Female Reference Interval					Male Reference Interval				
Age Group, y	Participants, N	Lower Limit	Upper Limit	Lower Limit 90% CI	Upper Limit 90% CI	Participants, N	Lower Limit	Upper Limit	Lower Limit 90% CI	Upper Limit 90% CI
60–69	274	168	363	163 to 174	352 to 375	240	151	300	147 to 156	290 to 309
70–79	248	168	351	163 to 173	339 to 363	199	133	293	127 to 140	284 to 302
≥80	126	162	358	155 to 170	339 to 377	88	121	304	112 to 131	286 to 322

**Table 3 jcm-09-02856-t003:** Indirect reference intervals for platelet counts stratified according to age and sex. Last value of each patient was taken. pU-LRL and pU-LRL, permissible uncertainty of estimated lower and upper reference limit. Collapsed reference interval for females without age stratification is 157 pU-LRL (147, 167) to 365 pU-URL (345, 385).

	Female Reference Interval					Male Reference Interval				
Age Group	Patients, N	Lower Limit	Upper Limit	pU-LRL	pU-URL	Patients, N	Lower Limit	Upper Limit	pU-LRL	pU-URL
60–69	13,721	163	368	152 to 173	348 to 389	10482	135	325	126 to 144	304 to 340
70–79	10,272	164	392	153 to 175	370 to 414	7419	128	325	119 to 137	306 to 344
≥80	6399	149	371	138 to 159	350 to 392	3394	112	330	104 to 121	310 to 351

**Table 4 jcm-09-02856-t004:** Frequencies of abnormal platelet counts in a population-based setting in the Principality of Liechtenstein using the conventional reference limit (150–350 × 10^9^/L).

Sex	Age, y	N	Abnormal, N	Abnormal, % (95% CI)	Thrombocytopenia, N	Thromboyctopenia, % (95% CI)	Thrombocytosis, N	Thrombocytosis, % (95% CI)
Female	60–69	3840	774	20.2 (18.9, 21.59)	141		557	14.5 (13.4, 15.7)
	70–79	3687	796	21.6 (20.3, 22.9)	201	5.5 (4.8, 6.2)	595	16.1 (15, 17.4)
	≥80	3672	1003	27.3 (25.9, 28.8)	373	10.2 (9.2, 11.2)	630	17.2 (16, 18.4)
	All	11,199	2573	23 (22.2, 23.8)	715	6.4 (5.9, 6.9)	1782	15.9 (15.2, 16.6)
Male	60–69	4995	870	17.4 (16.4, 18.5)	429	8.6 (7.8, 9.4)	441	8.8 (8.1, 9.6)
	70–79	4039	825	20.4 (19.2, 21.7)	489	12.1 (11.1, 13.1)	336	8.3 (7.5, 9.2)
	≥80	2567	780	30.4 (28.6, 32.2)	547	21.3 (19.8, 22.9)	233	9.1 (8, 10.3)
	All	11,601	2475	21.3 (20.6, 22.1)	1465	12.6 (12, 13.2)	1010	8.7 (8.2, 9.2)

**Table 5 jcm-09-02856-t005:** Frequencies of abnormal platelet counts in a population-based setting in the Principality of Liechtenstein, applying sex- and age-dependent reference intervals for males as evaluated in the present study.

Sex	Age, y	N	Abnormal, N	Abnormal, % (95% CI)	Thrombocytopenia, N	Thromboyctopenia, % (95% CI)	Thrombocytosis, N	Thrombocytosis, % (95% CI)
Female	60–69	3840	787	20.5 (19.2, 21.8)	251	6.5 (5.8, 7.4)	536	14.0 (12.9, 15.1)
	70–79	3687	869	23.6 (22.2, 25)	319	8.7 (7.8, 9.6)	550	14.9 (13.8, 16.1)
	≥80	3672	1,084	29.5 (28.1, 34)	488	13.3 (12.2, 14.4)	596	16.2 (15.1, 17.5)
	All	11,199	2,740	24.5 (23.7, 25.3)	1058	9.4 (8.9, 10)	1682	15.0 (14.4, 15.7)
Male	60–69	4995	1,309	26.2 (25, 27.4)	429	8.6 (7.8, 9.4)	880	17.6 (16.6, 18.7)
	70–79	4039	971	24.0 (22.7, 25.4)	295	7.3 (6.5, 8.1)	676	16.7 (15.6, 17.2)
	≥80	2567	696	27.1 (25.4, 28.9)	276	10.8 (9.6, 12)	420	16.4 (15, 17.8)
	All	11,601	2976	25.7 (24.9, 26.5)	1000	8.6 (8.1, 9.1)	1976	17.0 (16.4, 17.7)

**Table 6 jcm-09-02856-t006:** Systematic tabulation of direct evaluations of reference intervals for seniors. Median of LRL for females is 156 × 10^9^/L and for males it is 140 × 10^9^/L; median of URL for females is 396 × 10^9^/L and for males it is 367 × 10^9^/L.

Authors	Year	Sex Stratification	Age Stratification	Age Range	Sex	N	Reference Group	LRL (90% CI)	URL (90% CI)	Ref.
**Books and Standard References**										
Giles	1981	no	no	NA	Female/Male	NA	NA	150 (NA)	450 (NA)	[55]
Laurell	1997	no	no	NA	Female/Male	NA	NA	125 (NA)	340 (NA)	Cited in [48]
Kratz et al.	2004	no	no	NA	Female/Male	NA	NA	150 (NA)	350 (NA)	[21]
Williams 9th ed.	2017	no	no	NA	Female/Male	NA	NA	175 (NA)	450 (NA)	[47]
Wintrobes 14th ed.	2019	no	no	NA	Female/Male	NA	NA	177 (NA)	406 (NA)	[58]
**Original Papers**										
Hermann et al.	2020	yes	yes	60–99	Female	648	Prospectively assembled cohort of subjectively healthy participants in Switzerland with exclusion of individuals with disease	167 (164–170)	355 (347, 362)	**
				60–69	Male	240		151 (147–156)	300 (290, 309)	
				70–79	Male	199		133 (127–150)	293 (284–302)	
				80–94	Male	88		121 (112–131)	304 (286–322)	
Biino et al. *	2013	yes	yes	15–64	Female	16,358	Pooled analysis of 3 population-based studies in Italy with exclusion of individuals with disease	156 (153–158)	405 (401–410)	[49]
				65–100	Female	4835		140 (137–144)	379 (372–390)	
				15–64	Male	13,789		141 (140–144)	362 (358–365)	
				65–100	Male	4303		122 (119–126)	350 (343–360)	
Nah et al.	2018	yes	yes	12–99	Female	196,419	Checkups of individuals performed at 16 health promotion centers in 13 Korean cities with exclusion of individuals with disease	159 (NA)	367 (NA)	[51]
				60–75	Male	NA		140 (NA)	367 (NA)	
				76–99	Male	NA		126 (NA)	367 (NA)	
Giacomini et al.	2001	yes	yes	45–65	Female	50	Blood donors and healthy individuals presenting to laboratory for checkup in Italy	156 (NA)	351 (NA)	[50]
				45–65	Male	50		156 (NA)	300 (NA)	
				66–85	Female/Male	100		139 (NA)	363 (NA)	
Adeli et al.	2015	yes	no	27–79	Female	1390	Canadian Health Measures Survey (CHMS) with exclusion of sick participants	153 (138–169)	361 (348–375)	[20]
				27–79	Male	1490		152 (144–159)	324 (315–333)	
Troussard et al.	2013	yes	no	55–69	Female	4739	Periodic health assessment in 11 prevention and public health centers in western France, mainly Caucasian; exclusion of participants with disease	187 (NA)	420 (NA)	[19]
				60–64	Male	1747		191 (NA)	393 (NA)	
Park et al.	2016	yes	no	>60–?	Female	40	Healthy adults who received a general health examination in Korea	135 (NA)	326 (NA)	[18]
				>60–?	Male	51		125 (NA)	347 (NA)	
Helmersson-Karlqvist et al.	2016	yes	no	80	Female	228	Population-based study in Uppsala with exclusion of patients with diabetes mellitus and/or cardiovascular disease	156 (136–175)	426 (387–466)	[17]
				80	Male	181		116 (99–134)	399 (306–493)	
Nordin et al.	2004	yes	no	18–91	Female	960	Healthy laboratory staff from 102 medical laboratories in Scandinavia and their adult family members	165 (159–173)	387 (375–403)	[48]
				18–90	Male	866		145 (138–145)	348 (334–358)	
Tsang et al.	1998	yes	no	49–97	Female	1837	Population-based study in Sydney with exclusion of individuals with disease	163 (NA)	414 (NA)	[57]
				49–97	Male	1382		153 (NA)	382 (NA)	
Jernigan et al.	1980	No	no	64–94	Female/Male	73	Healthy participants at a periodic health exam	140 (NA)	440 (NA)	[24]
Janu et al.	2003	no	no	75–100	Female/Male	338	Community-dwelling cohort with random sampling in Sydney (including 38 nursing home residents)	150 (NA)	450 (NA)	[23]
Huber et al.	2006	no	no	75	Female/Male	119	Selection of apparently healthy individuals from a prospective population-based study in Vienna by exclusion of individuals with disease	114 (84–144)	452 (389–472)	[56]

LRL, lower reference limit; URL, upper reference limit; CI, confidence interval. NA, Not available. * 95% CI was used for reference limits. ** Present study.

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
