# Peer review of "Reference Intervals for Platelet Counts in the Elderly: Results from the Prospective SENIORLAB Study"

_jcm, 2020, doi:10.3390/jcm9092856_

Round 1
Reviewer 1 Report
Congratulations to the authors on this work. Very well written manuscript. The authors did a great job of explaining their methods leading to the conclusion.
Their data yielded a difference in normal platelet counts between elderly persons. Females have higher normal limits. At the same time, Males have age-dependent normal limits of normal platelet counts.
The authors used data from the prospective SENIORLAB study. The authors used data from about 1000 patients for the initial cohort (SENIORLAB study data). This data was then validated using data from routine clinical practice lab, including ~ 51K patients making those results reliable.
In ordered to determine the normal limits of platelets, data from patients with other causes of thrombocytopenia’s were excluded.
Their data yielded a difference in platelet counts between elderly persons. Females have higher normal limits. At the same time, males have age-dependent normal limits of platelets. For elderly males, normal platelet counts trend down with age.
A minor thing in the abstract the authors might consider:
"females have higher platelet counts then males."
Author Response
Congratulations to the authors on this work. Very well written manuscript. The authors did a great job of explaining their methods leading to the conclusion.
Their data yielded a difference in normal platelet counts between elderly persons. Females have higher normal limits. At the same time, Males have age-dependent normal limits of normal platelet counts.
The authors used data from the prospective SENIORLAB study. The authors used data from about 1000 patients for the initial cohort (SENIORLAB study data). This data was then validated using data from routine clinical practice lab, including ~ 51K patients making those results reliable.
In ordered to determine the normal limits of platelets, data from patients with other causes of thrombocytopenia’s were excluded.
Their data yielded a difference in platelet counts between elderly persons. Females have higher normal limits. At the same time, males have age-dependent normal limits of platelets. For elderly males, normal platelet counts trend down with age.
Thank you for your review and appreciating our work.
- A minor thing in the abstract the authors might consider: "females have higher platelet counts then males."
The sentence now reads: “In conclusion, females have higher platelet counts than males.”
Reviewer 2 Report
This manuscript reports on an extensive study aiming at defining age-and-sex-dependent reference limits for platelet counts in senior individuals. The work was performed in Switzerland and Liechtenstein from several cohorts of healthy individuals, well characterized for biological parameters.
The introduction is slightly overlong and could be more concise, although it sets properly the context of the study and its potential importance for early detection of platelet anomalies in senior populations.
A better definition of the “direct” and “indirect” studies should be provided. It is assumed from reading the manuscript that “direct” refers to SENIORLAB while “indirect” comes from analyses of people outside this trial, yet all tested in a single lab and possibly several times over the observation period. This deserves more explanation and perhaps another choice of words than “direct” and “indirect”.
The three groups mentioned in the statistics paragraph should be defined in methods or earlier in this section.
The results section is clear, save for the distinction between, “direct” and “indirect” mentioned above.
In Figure 3, it could be interesting to also indicate the “new” reference intervals. Showing all the dots would probably make the figure illegible but perhaps an additional figure in supplemental material could show the difference of cases above or below the “classical” and ”new” ranges for each age group.
The notion of estrogen levels assay suddenly appears in the discussion. If this comes from the published data of SENIORLAB, it should be mentioned. Otherwise, this should appear in Methods, especially since this criterion is part of the conclusion of the authors. The lower level of platelets in senior males is quite obvious from studies shown in Table 4 and should appear as a confirmation in the conclusion. More important, since a diagnosis of myeloproliferative neoplasm calls for the repeated observation of thrombocytosis, could the authors evaluate for how many individuals this diagnosis was delayed by use of the classical reference range? Similarly, how many cases of thrombocytopenia would have been diagnosed in excess, especially in males?
Minor remarks
Than instead of then in the conclusion of the abstract
Haemolysis instead of hamolysis in the third paragraph of the introduction.
Mention Vitamin B12 deficiency as for iron and folates.
One instance too many of “of the entire population” in the description of the Liechtenstein cohort.
Disorder not disorders in the caption of Figure 1.
Figures in tables should be homogenized (use or not of the “,” thousand mark)
MPV should be explained
